# The Influence of Emotional Intelligence on Quality of Life in Patients Undergoing Chronic Hemodialysis Focused on Age and Gender

**DOI:** 10.3390/bs14030220

**Published:** 2024-03-08

**Authors:** Afra Masià-Plana, Miquel Sitjar-Suñer, Susana Mantas-Jiménez, Rosa Suñer-Soler

**Affiliations:** 1Nursing Department, Faculty of Nursing, University of Girona, 17003 Girona, Spain; afra.masia@udg.edu (A.M.-P.); miquel.sitjar@udg.edu (M.S.-S.); rosa.sunyer@udg.edu (R.S.-S.); 2Research Group Health and Health Care, Nursing Department, University of Girona, 17003 Girona, Spain

**Keywords:** chronic kidney disease, hemodialysis, emotional intelligence, quality of life, gender differences, nursing

## Abstract

Emotional intelligence is a health predictor as it has an effect on quality of life, given that it affects a person’s capacity to use and regulate emotions and consequently has an influence on their physical and mental condition. The aim of this study is to analyze emotional intelligence, quality of life, and associated correlation factors in patients undergoing chronic hemodialysis, differentiating age and gender differences. A multicenter study of one hundred and thirty-eight hemodialysis patients was conducted using a quantitative, observational, and cross-sectional design. A battery of questionnaires, including Trait Meta-Mood Scale (TMMS-24), Kidney Disease Quality of Life—Short Form (KDQOL-SF), and an ad hoc sociodemographic questionnaire, were administered. The Chi-squared test, the Student’s *t*-test, and one-way ANOVA were used to compare categorical, independent, and continuous variables, respectively. A linear regression model was used to compare variables associated with quality of life. Gender differences existed when assessing the three dimensions of emotional intelligence: ranked by order, the highest scores in males were in clarity, repair, and attention. However, in females, the highest scores were in repair, clarity, and attention. Males had higher scores than females in emotional roles (*p* = 0.045). Younger participants had better scores in all physical functions (*p* = 0.001) and vitality (*p* = 0.038). Participants who had a higher level of education presented better physical function (*p* = 0.027) and higher scores in emotional wellbeing (*p* = 0.036). Finally, in females, emotional attention (*p* = 0.046) and repair (*p* = 0.000) were strongly associated with general health perception. The assessment of emotional intelligence considering gender differences in patients undergoing chronic hemodialysis can be an indicator of quality of life, including for use in health interventions.

## 1. Introduction

### 1.1. Chronic Kidney Disease and Its Impact

Chronic kidney disease (CKD) is an incurable serious illness that is recognized as a major public health concern worldwide. Its estimated prevalence is between 11.7% and 15.1% in the world population [1,2]; the number of individuals suffering from it is growing globally, and it is expected that the number of patients who will require renal replacement therapy in the coming years will increase [3,4,5,6]. The global all-age mortality rate from CKD increased by 41.5% between 1990 and 2017 [7].

The most widely used therapy worldwide for CKD is hemodialysis (HD) [8], which entails absolute life dependency on a blood filtering machine. The patient undergoing HD therapy usually suffers from stressful and threatening situations throughout their life, which often results in disability as well as physical and psychological suffering [9]. Moreover, there are multiple adverse events in patients who suffer from advanced chronic kidney disease, such as pain [10], and these can have a strong influence on patient counseling and medical decision-making [11]. Mortality in renal replacement therapy has decreased due to technological advances. However, the morbidity risk for these chronic kidney patients continues to increase due to the existence of negative mental and physical impacts that affect their health-related quality of life (QL) [12].

### 1.2. Emotional Intelligence

Research into the regulation and management of emotions and its potential implications for quality of life has aroused great interest in all areas [13,14]. Emotional intelligence (EI) is composed of three dimensions and can be defined as the ability to attend to (emotional attention), understand (emotional clarity), and modify (mood repair) emotional states [15], as well as the ability to regulate emotions to promote emotional and intellectual growth [16]. However, a high level of emotional attention interferes negatively with the positive effect of mood repair and personal growth if it is not complemented by a clear understanding of emotions [17].

Many studies show the role of emotional intelligence as a predictor of positive emotional states as well as wellbeing [18,19], and it has an effect on the quality of life of elderly people with chronic disease [20]. When measured as a trait, emotional intelligence is more strongly associated with health than when measured as an ability and can, therefore, be a possible health predictor [21].

Sex-related differences in emotion management are a reality given that sex differences in brain structure as well as psychological functioning do exist. Although females are better at recognizing emotions and expressing themselves more easily, males show greater responses to threatening cues, and this may reflect different behavioral response tendencies between males and females [22]. In order to increase the understanding of why males and females differ behaviorally and why many psychiatric diagnoses show unequal sex ratios, brain studies have been undertaken that show that sex- and gender-identity-related differences in humans’ effects come from experience, rather than causing variability in gender identity [23]. In a study where adult brains were analyzed, it was found that most brains did not show a consistent sex-typical pattern in all regions [24]. Similarly, the same study found that brain connectivity was never consistently male-typical or female-typical. This finding presents a challenge in correctly interpreting outcomes.

Emotional intelligence and quality of life studies in patients undergoing chronic hemodialysis therapy are scarce. However, in 2016, a positive correlation between both variables was proven. Two years later, a positive effect of an educational program in emotional intelligence on quality of life was observed in HD patients [25,26].

Considering the risks and consequences of emotional disorders among hemodialysis patients, the literature has highlighted the role of emotional intelligence as an adequate tool to face stressful situations and achieve emotional wellbeing [27].

### 1.3. Quality of Life in Patients Undergoing Chronic Hemodialysis Therapy

Health-related quality of life is a concept that is closely influenced by an individual’s perception of their physical and psychological state, level of independence, social relations, and their relationship with the environment [28].

Complications stemming from hemodialysis therapy that may interfere with different areas of quality of life are physical (loss of normal body function, sleep disorders, hemodialysis complications), psychological (pain, anxiety, depression, fatigue, self-image disturbance, decreased quality of life), and social (role change, perception of the disease, time in dialysis unit) [29,30,31].

Due to an increase in survival rates for patients with CKD, the outcome measures for dialysis therapy, such as quality of life, have become increasingly important not only as a quality assessment for health professionals but also for patient survival [32,33]. Studies show how low quality of life leads to higher mortality and a greater number of hospitalizations [12]. Moreover, there are some differences in the modality of dialysis in relation to quality of life outcomes [34].

The quality of life of hemodialysis patients is affected by emotion-focused coping strategies and cognitive processing, which can both be affected by socio-environmental and situational factors related to emotional intelligence [27].

The authors hypothesize that emotional intelligence could be a modulating factor of quality of life in people undergoing chronic hemodialysis therapy and that there are gender differences in this regard. The aims of the study are, firstly, we have analyzed emotional intelligence skills in relation to age and gender differences in patients undergoing chronic HD therapy. Secondly, we have studied the quality of life of HD patients by gender. Finally, we have described the correlation factors associated with emotional intelligence. It is expected that the findings of this study will draw more attention to the psychological distress of patients undergoing chronic HD therapy.

## 2. Materials and Methods

### 2.1. Study Design

A descriptive, observational, cross-sectional, analytical, multicenter study was conducted [35] to determine the factors that affect quality of life in chronic HD patients as well as differences by age and gender.

### 2.2. Participants

The study was directed at all 297 patients included in the chronic HD program of a region of Catalonia (Spain) in a convenience sample of four HD centers that presented similar characteristics. The inclusion criteria required patients to be older than 18 years of age, attend one of the four different HD units, and voluntarily agree to participate. Patients who suffered from mental disorders or a clinical condition at the time of data collection and those with language barriers were excluded. After the above criteria were applied, a total of 138 patients were eligible to respond to the self-assessment questionnaires (46.5%).

### 2.3. Data Collection

Initially, permission to contact the patients was requested from the clinics or hospital management as well as from the medical and nursing supervision members working in the hemodialysis units. We informed both the patients and staff of the hemodialysis units about the purpose of the study and the confidentiality of data and that participation was voluntary. Data collection was undertaken during the time of HD therapy, and paper-format questionnaires were handed out to be answered in situ. Some patients asked to be able to respond to them at home due to physical difficulties, such as needle insertions or not carrying reading glasses at the time of HD therapy, and these questionnaires were returned at the following sessions.

### 2.4. Variables

The independent variables used to explain variations in quality of life were sociodemographic variables (gender, age, living situation, and level of education). Emotional intelligence was a modulating variable for quality of life. The dependent variables were those related to quality of life, such as the physical and mental dimensions.

### 2.5. Instruments

The ad hoc questionnaire designed to gather sociodemographic data included the following variables: age and gender; level of studies (primary, secondary and university); living situation (alone, nursing home, living with family); employment status (employed, unemployed, retired, other situation); and monthly income (above or below EUR 645, based on the Spanish minimum standard salary in 2016).

We assessed emotional intelligence with the Trait Meta-Mood Scale (TMMS-24) [15] in its validated Spanish translation [36]. This is a self-report measure designed to assess an individual’s beliefs about his or her own emotional abilities. The scale addresses three key aspects of perceived emotional intelligence: Attention conveys to what extent individuals tend to observe and think about their feelings and moods, Clarity evaluates the understanding of one’s emotional states, and Repair refers to the individual’s beliefs about their ability to regulate their feelings. Specifically, the TMMS-24 is a 24-item Likert-type scale on which participants are required to rate the extent to which they agree with each item on a 5-point scale from 1 = strongly disagree to 5 = strongly agree. The scale appeared to have adequate psychometric characteristics with high reliability (Cronbach Alpha: perception α = 0.90; clarity, α = 0.90, repair α = 0.86) (test–retest reliability: perception = 0.60, clarity = 0.70, repair = 0.83) [37]. The score for the different factors was obtained from the sum of the set of items on each scale. Items 1–8, 9–16, and 17–24 evaluated emotional attention, clarity, and repair correlatively. Based on these scores, participants were classified as: “needs to improve”, “adequate”, or “excellent” in emotional clarity/repair and “needs to improve”, “adequate” and “too much” in emotional attention. This classification was made using different cut-off points for males and females as, according to the authors, there are gender differences in the handling of emotions.

Quality of life was measured with the Kidney Disease Quality of Life—Short Form questionnaire (KDQOL-SF) [38]. This consists of the generic 36-Item Short Form Health Survey (SF-36) [39] together with 11 multi-item scales focusing on quality of life issues that are specific to patients with kidney disease. The generic survey consists of 8 scales that can provide two summative scores: the physical and mental summative index. The internal consistency of the generic part is strong, with alpha values ranging from 0.78 to 0.92 [40]. The questionnaire is completed with 44 items specific to kidney disease distributed on 11 scales. For the present research, one item was removed since it only referred to patients undergoing peritoneal dialysis. The specific part of the questionnaire had a high internal consistency with a Cronbach’s alpha of α = 0.80 (with 2 exceptions: α = 0.68 in cognitive function and α = 0.61 as social interaction). The average values of the objective scales oscillated between 25.6 (DT = 37.82) for work and 79.11 (DT = 19.75) for cognitive function in the percentage of the possible total score (0–100). This instrument is the most widely used internationally as it assesses patients with kidney failure holistically [41].

### 2.6. Statistical Analysis

Statistical analysis was performed using Windows SPSS software version 25.0 for IBM. Continuous variables were described as the mean and standard deviation or the median and interquartile range, according to their probability distribution. Continuous variables were compared with the Student’s *t*-test or with a one-way ANOVA and the categorical variables were compared with the Chi-squared test. A multiple linear regression model (enter method) was carried out to study the factors related to quality of life. The variables accepted for the multiple linear regression analysis were those associated with emotional intelligence in the bivariant analysis. In all cases, a *p*-value < 0.05 was considered to be statistically significant.

### 2.7. Ethical Considerations

This research study is compliant with the fundamental ethical principles that govern the conduct of research. The anonymity of the participants was guaranteed at all times, and the informed consent form was signed by all participants prior to the start of data collection. The study was approved by the Clinical Research Ethics Committees of the area of influence (acceptance number: 2014.016 date: 24 February 2014 act #3).

## 3. Results

### 3.1. Study Sample Characteristics

The demographics and clinical characteristics of the participants are set out in Table 1. Nine out of ten participants lived with family and were retired, in a greater proportion in males than in females (*p* = 0.009). In terms of household income, it should be highlighted that more than two thirds of males had higher incomes than females (*p* = 0.029).

The hours of hemodialysis per session in males and females were 3.9 and 3.8, respectively (*p* = 0.004). The number of hospitalizations over the previous 3 months was 0.16 in males and 0.08 in females (*p* = 0.104).

### 3.2. Emotional Intelligence in Chronic Hemodialysis Patients

Table 2 shows the descriptive values of the three dimensions of emotional intelligence, split by gender, because males and females have different reference values. The highest scores in men were in clarity followed by repair and finally the attention dimension. In women, the highest scores were firstly in repair followed by clarity and finally attention.

When comparing emotional intelligence by age groups (one-way ANOVA test), we observed that in men, the attention dimension was higher in participants older than 64 years old, mean = 24.1 (7.5), and clarity, mean = 29 (6.8) (*p* = 0.818 and *p* = 0.926, respectively). On the other hand, in terms of repair, the highest scores corresponded to 46- to 65 year-old participants, mean = 28.3 (8.8), without statistical differences between men by age groups (*p* = 0.444).

In women, the 25- to 45-year-old age group presented the highest scores in attention, mean = 29 (9.3) (*p* = 0.082), and clarity, mean = 31.3 (74) (*p* = 0.208). In the repair dimension, the groups between 25 to 45 and 46 to 65 years of age had similar scores of 30.3 (7.8) and 30.6 (6.4), respectively, without statistical differences between women by age groups (*p* = 0.091).

Figure 1 and Figure 2 show the classification of emotional intelligence in males and females. As we can observe in Figure 1, the largest proportion of male participants scoring “Excellent” were in the clarity dimension. Of those scoring “Adequate”, the largest percentage was in the repair dimension. In women (Figure 2), however, the largest proportion of participants scoring “Excellent” was in the repair dimension followed by clarity, and a high proportion of those in attention were in “Needs to improve” (67.3).

### 3.3. Quality of Life in Chronic Hemodialysis Patients

A general overview of the results obtained regarding the physical dimensions of quality of life shows that scores were low. The four physical dimensions of quality of life are shown in Table 3. There were significative differences in age groups where younger participants had a better score in all the physical functions than older participants (*p* = 0.001). Moreover, those who had a higher level of education presented better physical function (*p* = 0.027).

With regard to the mental dimensions of quality of life (Table 4), men presented higher scores than women in the emotional role (*p* = 0.045). Younger participants had higher scores in vitality (*p* = 0.038). Finally, participants who had university studies had higher scores in emotional wellbeing (*p* = 0.036). 

In the multiple linear regression model used to study the variables associated with the perception of general health, no sociodemographic variables were found to be related to the three dimensions of emotional intelligence in men. However, attention (*p* = 0.046) and repair (*p* < 0.01) were found to be strongly associated with the perception of general health in women (Table 5 and Table 6).

## 4. Discussion

This study analyzed emotional intelligence skills and quality of life in a sample of 138 patients undergoing chronic hemodialysis therapy. With regard to emotional intelligence, we observed major differences in the dimensions of emotional attention and repair. More women than men (+33.3%) had low attention. However, 27.2% more men than women had adequate attention, and finally, 5.3% more men than women paid too much attention to emotions. The results also showed that 9% more women than men had excellent emotional repair. These contradicting results could be explained by the characteristics of the different emotional stages that chronic hemodialysis therapy represents. This contradicts other results reporting higher emotional intelligence skills in women in a healthy population [42]. These differing patterns constitute an indication of great sociodemographic diversity but could also be partially explained by young women in the progesterone phase of their menstrual cycle performing better in EI [43], which is a potential confounder that we did not control. Pardeler et al. (2018) argued that cognitive abilities need to be considered when assessing emotional intelligence in healthy individuals as gender differences are present [44]. When taking an ability-based approach to the analysis of EI rather than just trait emotional intelligence, it also appears that women have higher scores than men [45].

Across most emotion-related outcomes, trait emotional intelligence tends to be a stronger predictor, and consequently, O’Connor et al. (2019) suggest that new users of emotional intelligence should consider using a trait-based measure before assessing alternatives [14]. When using self-report measures, such as TMMS-24, gender differences in results are shown. Given that men perceive themselves as being more emotionally intelligent than women, these measures demonstrate levels of emotional attention that are too high [46].

It should be taken into account that negative emotions can lead to excessively high emotional attention and clarity, which can enhance a state of anxiety. For this reason, it is necessary to achieve a balanced level of emotional intelligence. Paying excessive attention to emotions can have detrimental effects, reducing the ability to regulate emotions [47]. Multiple aspects of intelligence need to be controlled when assessing emotional intelligence for the prediction of health outcomes [45].

When studying emotional intelligence by age, our results were similar to those found in the literature [48] as the ability to repair increased with age. These results could be related to life experience as well as better emotional control.

The existence of a positive correlation between emotional intelligence and quality of life in chronic hemodialysis patients was shown in 2016 using a different instrument from TTMS-24 [25]. Other studies showed no sex differences in terms of quality of life and emotional intelligence, marital status, and educational levels [49].

As for training in emotional intelligence, studies have shown excellent results in all dimensions. In a comparative study, significant correlations were found in the intervention group, giving statistically significant overall results: 42.00 ± 10.22 before the intervention and 58.24 ± 8.66 post-intervention. Emotional intelligence training has shown excellent results in improving aspects of quality of life, including reducing anxiety scores in HD patients [26,50]. Similar results have been achieved in breast cancer patients through physical activity and psychosocial interventions [51].

Our results regarding quality of life by gender and age vary slightly from others in the literature, as a study in Jordan on participants undergoing hemodialysis therapy showed that men and younger participants had a better mental quality of life than women participants in general [52]. A systematic review by Yapa [53], where quality of life and symptoms experienced by patients who have CKD and were not on dialysis were analyzed, concluded that quality of life decreased when symptoms increased. Thus, evidence on how and which symptoms change over time was inconclusive. One possibility could be due to gender-related biological factors as well as different lifestyles, socialization, and cultural norms [23]. Moreover, quality of life is one of the main concerns of policymakers and public health planners in the community as well as being an indicator of quality care. We aim to contribute to this knowledge with the present research [54].

There is a large body of research studying quality of life in patients who suffer other chronic conditions and the results are broadly similar. For example, Yalcin et al. (2008) demonstrated that higher emotional intelligence levels were related to a better quality of life and the general wellbeing of people with diabetes [55]. Moreover, the way that women with breast cancer regulate their emotions influences their quality of life and enhances disease adaptation [56]. Emotional intelligence studies in patients who suffer from chronic obstructive pulmonary disease (COPD) showed an association with all domains of quality of life regardless of age [57]. This suggests that, since emotional intelligence is a trainable skill, there is an opportunity to convert this knowledge into management programs in order to improve the quality of life and wellbeing of such patients. In this respect, a strong association between general health and the emotional intelligence of women has also been observed in the present study. These gender differences could be explained by different responses in processing emotional signals [22] and the interrelation between cognitive abilities and the understanding of the emotions [42].

Understanding that hemodialysis is a complex and expensive treatment, healthcare providers need to accept responsibility not only for the impact of the treatment but also for the environmental effects [58]. There are two key issues that require attention: the impact of environmental change on kidney health and the impact on the environment on the care of patients with kidney disease [59]. Healthcare providers can contribute to the sustainability of kidney care by empowering the patient to cope and manage the disease independently and by creating advanced care planning starting early in people’s lives with a broad scope and refine to include more specific limitations or certain medical procedures [60].

The use of patient-reported outcome measures (PROMs) is widespread worldwide as a means of collecting information on health outcomes directly from patients, including information about symptoms, health-related quality of life, and functional status [61]. In the present research, self-reported questionnaires were used in order to collect all data. However, we believe that additionally, PROMs instruments can provide an integral analysis as part of an analysis of quality of life [62]. By doing so, we can quantify the health status of patients and the impact of the disease so as to deliver the best effective care [63].

This research is not without limitations. Firstly, the results obtained are limited to the patients included in the chronic hemodialysis program of a region of Spain (Girona). Although the findings can be generalized to samples of similar characteristics, we highlight the need for further investigation with broader population groups, which will help us to better understand the exact significance of the variables studied here. Secondly, the use of self-report measures is prone to recall bias, and the cross-sectional design has not allowed causal associations or the direction of the associations to be studied. Thirdly, a study of comorbidities and cultural and social factors may have shed further light on the participants’ quality of life, emotional intelligence, and the management of their emotions. Fourthly, since most questionnaires were completed during HD therapy, the presence of medical staff and other patients may have influenced the objectivity of the responses and threatened data validity. Finally, we used the term gender; however, there are studies that indicate that differences exist between the sex phenotype of a person and the gender self-identity, and these can sometimes be different within the same person [23]. We believe that in future research on emotional intelligence, sex differences, and gender identity should be considered separately.

This study has strengthened existing evidence that chronic HD therapy can have a detrimental mental health effect on CKD patients. It has been proven that these negative health effects have an impact on patients’ wellbeing and quality of life and require effective measures in nursing practice to be developed and implemented. Given the results we have obtained and the fact that emotional intelligence improves a person’s ability to manage their emotions, both individual and group programs for hemodialysis patients could be proposed, taking into account personal characteristics, including sex and environment.

Currently available interventions to favor the development of EI should improve adaptive strategies and reduce negative moods. This involves training skills through practical activities, such as emotional reflection exercises, emotional introspection, taking consciousness of one’s own emotions, and the acquisition of skills related to empathy and assertiveness. All of this is carried out with the aim of better regulating one’s own emotions, improving quality of life, and reducing related comorbidities, especially emotional distress [64]. Furthermore, given that the burden of the disease associated with chronicity has an influence on a deficiency in EI [65], satisfaction with life should be promoted through programs based on the use of adaptive coping strategies, which reduce the use of maladaptive strategies, despair, and symptoms of depression [66].

A holistic view of CKD patients should be adopted by nephrology nurses in order to consider and care for different aspects of their suffering, including emotion management. Emotional intelligence is a promising protective factor for biological and psychological variables in populations who suffer a chronic condition [67]. Likewise, renal association guidelines currently recommend health-related quality of life to be monitored in patients undergoing renal replacement therapy [68].

We suggest that clinical practitioners treating chronic hemodialysis patients should assess levels of emotional attention, especially in women. If instability is detected, a complete psycho-emotional intervention should be conducted. Nurses have the capacity to establish a helping relationship and have key resources at their disposal to do so, for example, through empowering hemodialysis patients to enhance their coping strategies [69]. Research is currently engaged in the active investigation of better tools, programs, and ultimately better outcomes for these patients [70].

The results of the present study may be applicable to other healthcare settings that provide hemodialysis treatment, always taking into account the importance of monitoring patients’ moods in order to make an early intervention in the event of any emotional imbalance.

## 5. Conclusions

The assessment of emotional intelligence in patients undergoing chronic hemodialysis therapy can be an indicator of clinical outcomes and improve health interventions, as well as being a tool for evaluating the therapy carried out. Improving quality of life and subsequently enhancing emotional intelligence may prevent irreversible complications in patients undergoing chronic HD therapy.

Nurse managers need to guide nurses on the application of emotional intelligence skills in daily practice with the aim of providing holistic patient care, obtaining positive trait emotional intelligence outcomes and preventing psychological disorders.

## Figures and Tables

**Figure 1 behavsci-14-00220-f001:**
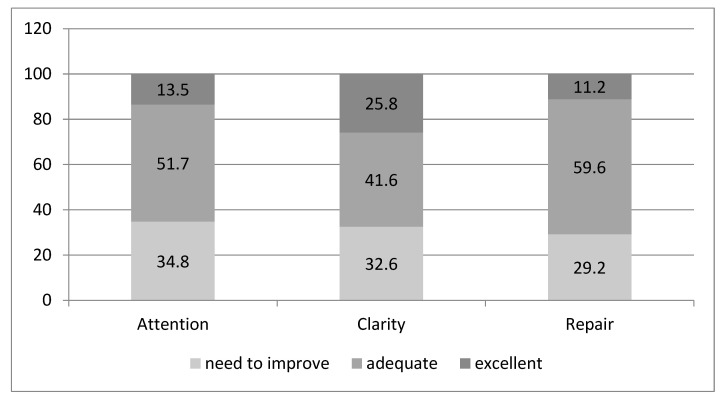
Level of emotional intelligence in men (%).

**Figure 2 behavsci-14-00220-f002:**
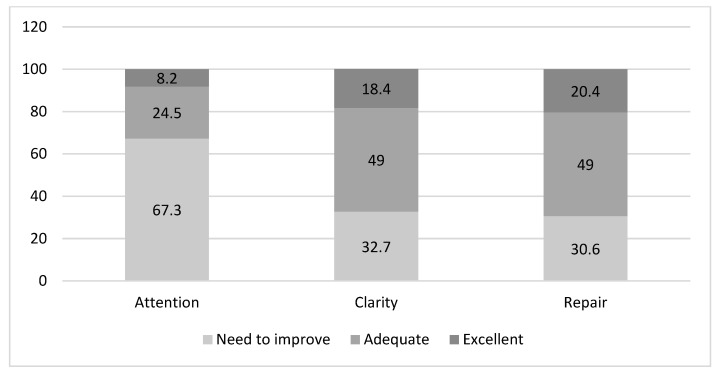
Level of emotional intelligence in women (%).

**Table 1 behavsci-14-00220-t001:** Characteristics for the overall sample and by gender.

	Global *n*: 138	Men *n*: 89	Women *n*: 49	*p*
**Age (SD)**	65.2 (15.5)	65.2 (15)	65.3 (16.4)	0.964
**Education level**				
Primary Secondary University	106 (76.8) 25 (18.1) 7 (5.1)	67 (75.3) 17 (19.1) 5 (5.6)	39 (79.6) 8 (16.3) 2 (4.1)	0.836
**Living situation**				
Alone With family Nursing home	18 (13) 118 (85.6) 2 (1.4)	8 (9) 80 (90) 1 (1)	10 (20.4) 38 (77.6) 1 (2)	0.142
**Clinical characteristics**				
Pain (Visual Analogue scale)	3.6 (3)	3.1 (2.9)	4.4 (3.1)	0.015
Hemodialysis time (months)	55.2 (56.5)	54.4 (60.7)	56.5 (48.5)	0.831

Categorical variables are described as absolute frequencies and percentages, and continuous variables are described as means (standard deviation). The Chi-squared test was used to compare categorical variables, and the Student’s *t*-test was used to compare independent groups.

**Table 2 behavsci-14-00220-t002:** Description of emotional intelligence by gender.

	Male		Female	
	Mean (Standard Deviation)	Median (Interquartile)	Mean (Standard Deviation)	Median (Interquartile)
**Attention**	23.8 (7.466)	24.00 [17–30]	22.29 (8.319)	21.00 [16.50–29.50]
**Clarity**	28.80 (7.478)	27.00 [24–36]	26.59 (8.178)	25.00 [23.00–31.50]
**Repair**	27.29 (7.271)	29.00 [22–33]	27.35 (8.268)	28.00 [20.50–34.00]

**Table 3 behavsci-14-00220-t003:** Relationship between physical dimensions of quality of life and sociodemographic variables.

	Physical Function		Physical Role		Pain		General Health	
	Mean	*p*	Mean	*p*	Mean	*p*	Mean	*p*
**Gender**								
Male Female	52.13 (28.56) 46.00 (30.12)	0.120	42.97 (40.41) 44.38 (38.94)	0.422	57.27 (31.94) 53.62 (31.89)	0.261	37.58 (20.74) 36.83 (20.45)	0.419
**Age**								
25–45	75.52 (17.15)	<0.001	56.57 (38.94)	0.305	62.76 (27.87)	0.207	39.73 (17.27)	0.166
46–65	57.16 (27.14)		41.89 (38.66)		61.28 (31.70)		42.02 (20.29)	
>65	40.79 (27.99)		41.15 (40.51)		52.01 (32.52)		34.63 (21.15)	
**Living** **situation**								
Alone	57.22 (32.86)	0.199	38.88 (39.50)	0.478	55.13 (33.06)	0.242	38.88 (16.58)	0.348
With family	48.38 (28.53)		43.64 (40.19)		55.46 (31.69)		36.73 (21.09)	
Nursing homes	77.50 (10.60)		75.00 (0.00)		93.75 (8.83)		57.50 (17.67)	
**Level of** **education**								
Primary	46.32 (29.09)	0.027	39.85 (39.30)	0.122	53.58 (31.50)	0.160	36.42 (20.39)	0.604
Secondary	61.80 (27.07)		53.00 (39.07)		60.80 (30.10)		39.60 (22.68)	
University	62.85 (25.79)		64.28 (45.31)		75.00 (39.55)		42.85 (15.77)	

Results are shown with their mean and standard deviations. Continuous variables were compared with the Student’s *t*-test or with a one-way ANOVA.

**Table 4 behavsci-14-00220-t004:** Relationship between mental dimensions of quality of life and sociodemographic variables.

	Emotional Wellbeing		Emotional Role		Social Function		Vitality	
	Mean	*p*	Mean	*p*	Mean	*p*	Mean	*p*
**Gender**								
Male	64.51 (23.00)	0.235	76.40 (37.42)	0.045	66.03 (27.39)	0.173	49.32 (24.27)	0.187
Female	61.38 (26.52)		64.11 (45.33)		61.22 (30.85)		45.30 (27.22)	
**Age**								
25–45	68.42 (24.54)	0.463	73.68 (39.40)	0.916	71.44 (30.22)	0.447	60.26 (22.20)	0.038
46–65	65.18 (23.41)		73.87 (38.59)		65.20 (24.49)		49.72 (23.83)	
>65	61.43 (24.63)		70.83 (42.27)		62.28 (30.04)		44.20 (25.93)	
**Living** **situation**								
Alone	62.00 (23.61)	0.679	79.62 (34.56)	0.693	65.97 (26.36)	0.964	44.44 (26.00)	0.537
With family	63.37 (24.58)		70.97 (41.64)		64.11 (29.14)		48.13 (25.41)	
Nursing homes	78.00 (2.82)		66.66 (47.14)		62.50 (35.35)		65.00 (07.07)	
**Level of** **education**								
Primary	60.83 (24.17)	0.036	69.26 (41.77)	0.346	61.93 (28.23)	0.182	46.41 (25.78)	0.143
Secondary	69.19 (24.11)		81.33 (36.10)		71.00 (29.69)		49.20 (23.34)	
University	81.71 (15.45)		80.95 (37.79)		76.78 (28.34)		65.71 (20.70)	

Results are shown with their mean and standard deviations. Continuous variables were compared with the Student’s *t*-test or with a one-way ANOVA.

**Table 5 behavsci-14-00220-t005:** Multiple linear regression model: variables associated with quality of life with general health in men.

Coefficients						
	Non Standardized Coefficients		Standardized Coefficients	*t*	Sig.	95.0% CI for B
	B	Std. Error	β			
**Constant**	80.449	76.904		1.046	0.299	−72.511 to 233.409
**Age**	0.590	0.744	0.087	0.793	0.430	−0.890 to 2.070
**Level of education**	10.405	19.832	0.058	0.525	0.601	−29.041 to 49.850
**Attention**	−1.762	1.659	−0.128	−1.062	0.291	−5.062 to 1.538
**Clarity**	−0.253	2.057	−0.018	−0.123	0.903	−4.345 to 3.839
**Repair**	3.805	2.200	0.270	1.730	0.087	−0.570 to 8.180

Dependent variable: general health; CI: confidence interval; R2 = 0.06.

**Table 6 behavsci-14-00220-t006:** Multiple linear regression model: variables associated with quality of life with general health in women.

Coefficients						
	Non Standardized Coefficients		Standardized Coefficients	*t*	Sig.	95.0% CI for B
	B	Std. Error	β			
**Constant**	242.504	96.294		2.518	0.016	48.309 to 436.699
**Age**	−1.666	0.866	−0.266	−1.923	0.061	−3.414 to 0.081
**Level of education**	−6.354	27.897	−0.032	−0.228	0.821	−62.614 to 49.905
**Attention**	−3.730	1.816	−0.301	−2.055	0.046	−7.392 to −0.069
**Clarity**	−2.004	2.085	−0.159	−0.961	0.342	−6.209 to 2.201
**Repair**	7.163	2.025	0.574	3.538	<0.001	3.080 to 11.247

Dependent variable = general health; CI: confidence interval; R2 = 0.31.

## Data Availability

The data presented in this study are available on request from the corresponding author.

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
