# Peer review of "The Influence of Emotional Intelligence on Quality of Life in Patients Undergoing Chronic Hemodialysis Focused on Age and Gender"

_behavsci, 2024, doi:10.3390/bs14030220_

Round 1

Reviewer 1 Report

Comments and Suggestions for Authors

The article provides valuable insights into the influence of emotional intelligence on the quality of life of chronic hemodialysis patients. The use of the Kidney Disease Quality of Life questionnaire and statistical analysis using SPSS software demonstrates a robust methodology. The findings regarding the association between emotional intelligence, sociodemographic variables, and the perception of general health in hemodialysis patients are significant.

However, the study could benefit from a more detailed discussion of the limitations mentioned and their potential impact on the findings. Some additional limitations of the study could include the potential impact of comorbidities on the emotional intelligence and quality of life of hemodialysis patients, as well as the potential influence of cultural and social factors on the perception and expression of emotions in the study population. Furthermore, the study's reliance on a convenience sample from a specific region may limit the generalizability of the findings to other populations with different sociodemographic and cultural characteristics. These limitations could have implications for the external validity and applicability of the study's findings to broader patient populations and healthcare settings.

Additionally, the article would be strengthened by further exploration of the implications for patient care and specific recommendations for interventions based on the emotional intelligence findings.

Reviewer 2 Report

Comments and Suggestions for Authors

Reviewer 3 Report

Comments and Suggestions for Authors

In this paper the authors aim to assess how emotional intelligence can be predictive of quality of life in patients undergoing kidney dialysis, which in turn is key to prevent complications as a form of tertiary prevention.

The paper is well-structured, consistent and clear, although it needs deep English revision to make it more fluent and clarify some prepositions.

Although I personally believe in the use of patient-reported outcomes and the intrinsic connection between psychosocial health, physical health and prevention, I wonder what could be the impact -  as well as the originality - of understanding how quality of life is reduced by patients undergoing kidney dialysis.

The authors write that some interventions (including nursing care) can help these patients cope better with therapy, but considering the increasing relevance of kidney failure and chronic diseases all over the world, how can it be sustainable?

It would be interesting to hear the opinion of the authors on this point, in the discussion.

Comments on the Quality of English Language

Extensive English revision can help the paper be written more fluent and clarify some points.
